# Dynamic Simulation Analysis and Optimization of Green Ammonia Production Process under Transition State

**Wu Deng [1], Chao Huang [2], Xiayang Li [3], Huan Zhang [3] and Yiyang Dai [2],***

[1] Puguang Economic Development Zone, Dazhou 635000, China
[2] Process Systems Engineering Research Group, College of Chemical Engineering, Sichuan University, Chengdu 610000, China
[3] Research and Development Department, Tsingyun Intelligence Technology Co., Ltd., Chengdu 610000, China
* Correspondence: daiyy@scu.edu.cn

**Abstract:** Ammonia is an important chemical raw material and the main hydrogen energy carrier. In the context of "carbon neutrality", green ammonia produced using renewable energy is cleaner and produces less carbon than traditional ammonia production. Raw hydrogen dynamically fluctuates during green ammonia production because it is affected by the instability and intermittency of renewable energy; the green ammonia production process has frequent variable working conditions to take into account. Therefore, studying the transition state process of green ammonia is critical to the processing device's stable operation. In this study, a natural gas ammonia production process was modified using green ammonia, and steady-state and dynamic models were established using UniSim. The model was calibrated using actual factory data to ensure the model's reliability. Based on the steady-state model, hydrogen feed flow disturbance was added to the dynamic model to simulate the transition state process under variable working conditions. The change in system energy consumption in the transition state process was analyzed based on the data analysis method. The proportional-integral-derivative (PID) parameter optimization method was developed to optimize energy consumption under variable conditions of green ammonia's production process. Based on this method, process control parameters were adjusted to shorten fluctuation time and reduce energy consumption.

**Keywords:** green ammonia; UniSim; transition state; dynamic simulation; process control



## 1. Introduction

Ammonia is necessary for food production and as an energy carrier and chemical raw material. The most common industrial ammonia synthetization method is the Haber–Bosch (HB) process. The Haber–Bosch process is an energy-intensive industrial ammonia synthesis process. To make N≡N(945 kJ·kmol$^{-1}$) fracture, the high temperature and high pressure conditions (673 K~873 K and 20 MPa~40 MPa, respectively) required by the reaction should be maintained [1]. Conventional ammonia synthesis processes consume 2% of global energy output and account for 1% of global greenhouse gas emissions [2]. Therefore, in the context of carbon neutrality, the green transformation of the ammonia synthesis process is particularly important to save energy, reduce consumption and greenhouse gas emissions and achieve renewable and sustainable ammonia synthesis.

Wang [3] summarized several green ammonia processes based on existing studies, including thermal catalysis, electrocatalysis, photocatalysis, and chemical cyclization [4,5]. Thermal catalytic synthesis of ammonia has been applied in industry [6]; the remaining green ammonia processes are in the research and development stage, so this paper will not discuss them. The thermal catalysis method is an improvement over the HB process. Using a Fe- or Ru- based catalyst, reaction temperature can be reduced to 300–450 °C, pressure can be reduced to 4~20 MPa, reaction conditions can be improved, and energy saving and consumption reduction can be achieved.

In addition to saving energy and reducing consumption, the new ammonia production process requires decarbonization to achieve carbon neutrality [7]. Methane steam reforming and coal gasification are the primary methods used to produce raw hydrogen and $CO_2$, respectively, in the ammonia industry. To realize technological decarbonization, Guo [8] makes three recommendations: 1. Collect and store $CO_2$ generated during the process. 2. Use green hydrogen, produced using renewable energy, as a feedstock. 3. Adopt new production processes that use $H_2O$ as a raw material [9]. Given that the new process is in the research and development stage, that industrial application has not yet been verified, and economic considerations, the use of green hydrogen as a raw material is the optimal choice.

Green hydrogen produced via water electrolysis using renewable energy sources such as wind, light, and hydropower is low-carbon and environmentally friendly, and its production scale is constantly expanding, which indicates good prospects for large-scale application [10]. Hydrogen is difficult to store, explosive, and its transportation has many technical requirements. However, green hydrogen can be directly used in ammonia production. Using ammonia as a hydrogen energy carrier eliminates the problematic storage and transport of hydrogen [11]. Ammonia that uses green hydrogen as a raw material is called green ammonia because its manufacturing process has zero emissions.

To sum up, combined with the current mature technology, this study adopted the thermal catalytic method to transform traditional industrial HB ammonia synthesis and used green hydrogen as a raw material. Nitrogen was separated from air using pressure swing adsorption (PSA), which has the advantage of low cost and simple processing. Figure 1 illustrates the technical realization of the green ammonia process.

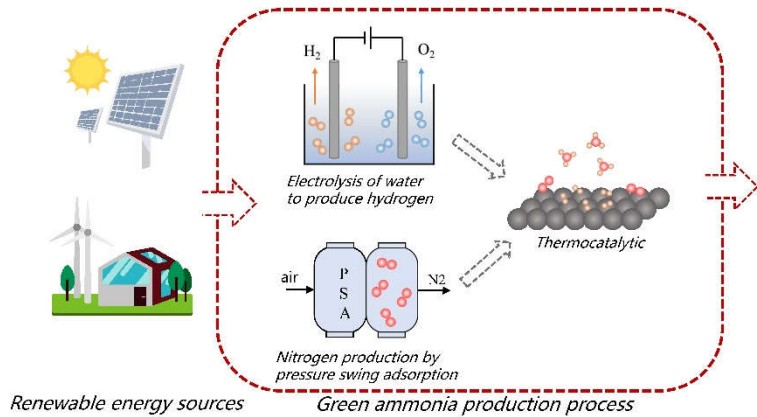

**Figure 1.** Green ammonia production process technical route.

This study divided the green ammonia production process into three steps:

1. Process simulation technology and UniSim software were used to design and build a green ammonia steady-state model, and actual plant data were used to reconcile the model and ensure model data conformed to the actual production process.
2. Based on the steady-state model, a dynamic model was developed to study green ammonia's transition state process.
3. Based on the dynamic model, a proportional-integral-derivative (PID) parameter optimization method, based on energy consumption optimization under variable conditions, was proposed. This dynamic process was optimized using the PID method to reduce the transition state's duration and energy consumption.

## 2. Green Ammonia Process

A synthetic ammonia plant uses natural gas reforming to produce hydrogen and traditional HB technology to produce ammonia. It uses a four-stage reactor (Kellogg Brown & Root, Houston, TX, USA) to synthesize ammonia, and its 100% load production is 62,539 kg $NH_3$/h. Figure 2 illustrates a simple process flow chart for green ammonia transformation.

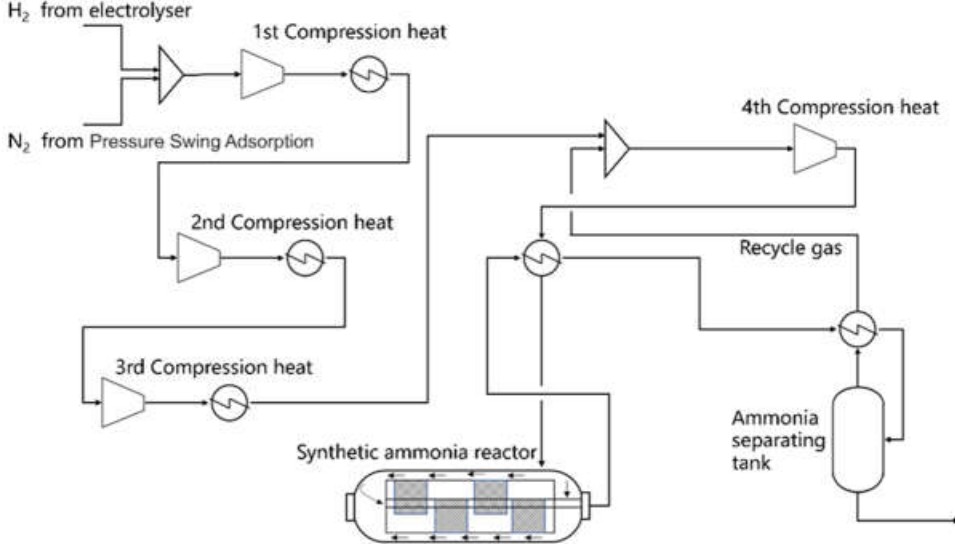

**Figure 2.** Green ammonia process flow chart.

The feedstock gas was composed of electrolytic device $H_2$ and air separation $N_2$, with an $H_2$:$N_2$ volume ratio of 3:1. The pressure of the raw gas from the water electrolysis device was limited by the water electrolysis working conditions, and the maximum pressure was not more than 2.0 MPa. Therefore, the syngas pressure needed to be increased, and the raw gas's pressure was raised to 14 MPa using four-stage compression, after which it entered the synthetic ammonia reactor (Kellogg Brown & Root, Houston, TX, USA). Products exported from the synthetic ammonia reactor were separated from ammonia products, and unreacted synthetic gas was recycled and mixed with fresh raw gas entering the synthetic ammonia reactor through a four-stage compressor.

At the same time, the water electrolysis hydrogen production unit's output was approximately 20 Nm$^3$/h; its ammonia output was calculated using an 8% conversion rate. According to the calculation results, the initial output was set as approximately 10,000 kg NH3/h under a 100% load.

Table 1 presents a technical comparison between the designed green ammonia process and the original process at 100% load.

**Table 1.** Technological comparison of the green ammonia process and the original process.

| Items | Original Process | Green Ammonia Process |
|---|---|---|
| Reactor temperature | 673 K~873 K | 623 K–723 K |
| Reactor pressure | More than 20 MPa | 14 MPa |
| Production | 62,539 kg NH$_3$/h | Approximately 10,000 kg NH$_3$/h |
| Hydrogen source | Natural gas steam reforming | Electrolysis of water |
| Hydrogen energy consumption | 41,077 kg Natural gas/h | 105~110 KW/h |
| Carbon emission | 84,662 kg CO$_2$/h | 0 kg CO$_2$/h |
| Catalyst | Osmium-based catalyst | Iron-based catalyst |

Next, a designed green ammonia process's chemical process modeling study was carried out. Vaccari [12] presented an offline simulation model developed using process simulation techniques that use analytical simplifications to overcome the burden linked to simulation costs. In the study of chemical modeling processes [13], many scholars have established energy efficiency models to analyze the operational energy consumption

of chemical units and to identify a production process's main energy-consuming equipment [14–17]. These works prove that it is feasible to use process simulation techniques to optimize energy efficiency and design.

## 3. UniSim Model of the Green Ammonia Process

The raw hydrogen used in the green ammonia process fluctuated because it was derived from renewable energy. This required frequently adjusting the production device's operational conditions as part of the transition state process. The transition state process's uncertainty was related to, and required for, the device's safe operation and production. Process dynamic simulation technology has been widely used to study transition state processes in chemical plants [18–22]. Therefore, this study used UniSim software [23,24] to establish a stable dynamic energy and mass balance model for the green ammonia process, which simulates the fluctuations of raw material hydrogen by adding disturbances and analyzes the green ammonia process's total energy consumption and its performance during the transition state process. The selection of process control parameters [25–30] is important for the smooth operation of the process. In the past, the selection of optimal PID control parameters mostly relied on experience and lacked systematic optimization ideas. Based on the established green ammonia process model and the dynamic optimization concept [31,32], we propose a new concept for PID control parameter selection, namely, a PID control parameter optimization method based on energy consumption optimization under variable conditions.

### 3.1. Steady-State Model

First, the steady-state model was built according to the process design results. The Peng–Robinson (PR) equation of state [33] was used as the physical property analysis method. The equation of state is suitable for the calculation of gas–liquid thermodynamic properties and gas–liquid equilibrium of a variety of non-polar or weakly polar substances; therefore, it is also suitable for the ammonia synthesis process. A synthetic ammonia reactor has a complex structure. As the synthetic ammonia reaction is exothermic, heat needs to be removed to maintain a stable temperature in the reactor, and the reaction conversion rate is low. To control the reactor's internal temperature and simulate the reactor's heat transfer behavior, a four-stage advection reactor and interstage heat transfer were used to simulate the ammonia tower structure.

The steady-state model of the UniSim process (with control points removed) was established according to the process flow charts illustrated in Figure 3.

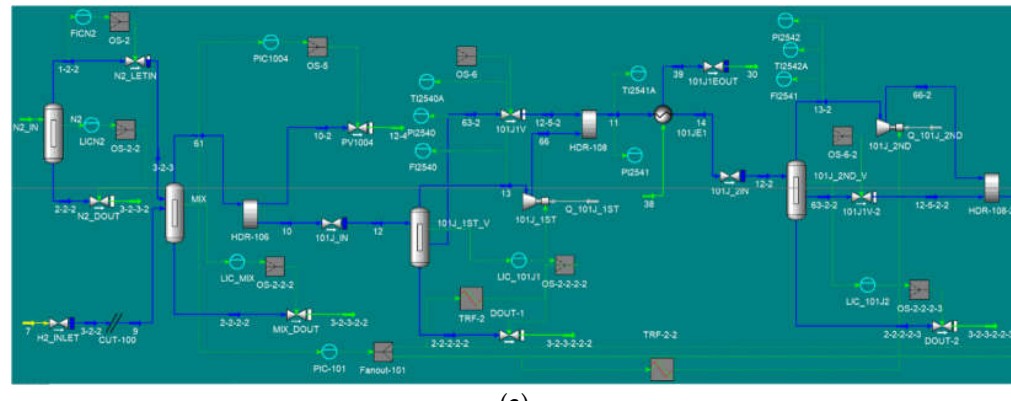

(**a**)

**Figure 3.** *Cont.*

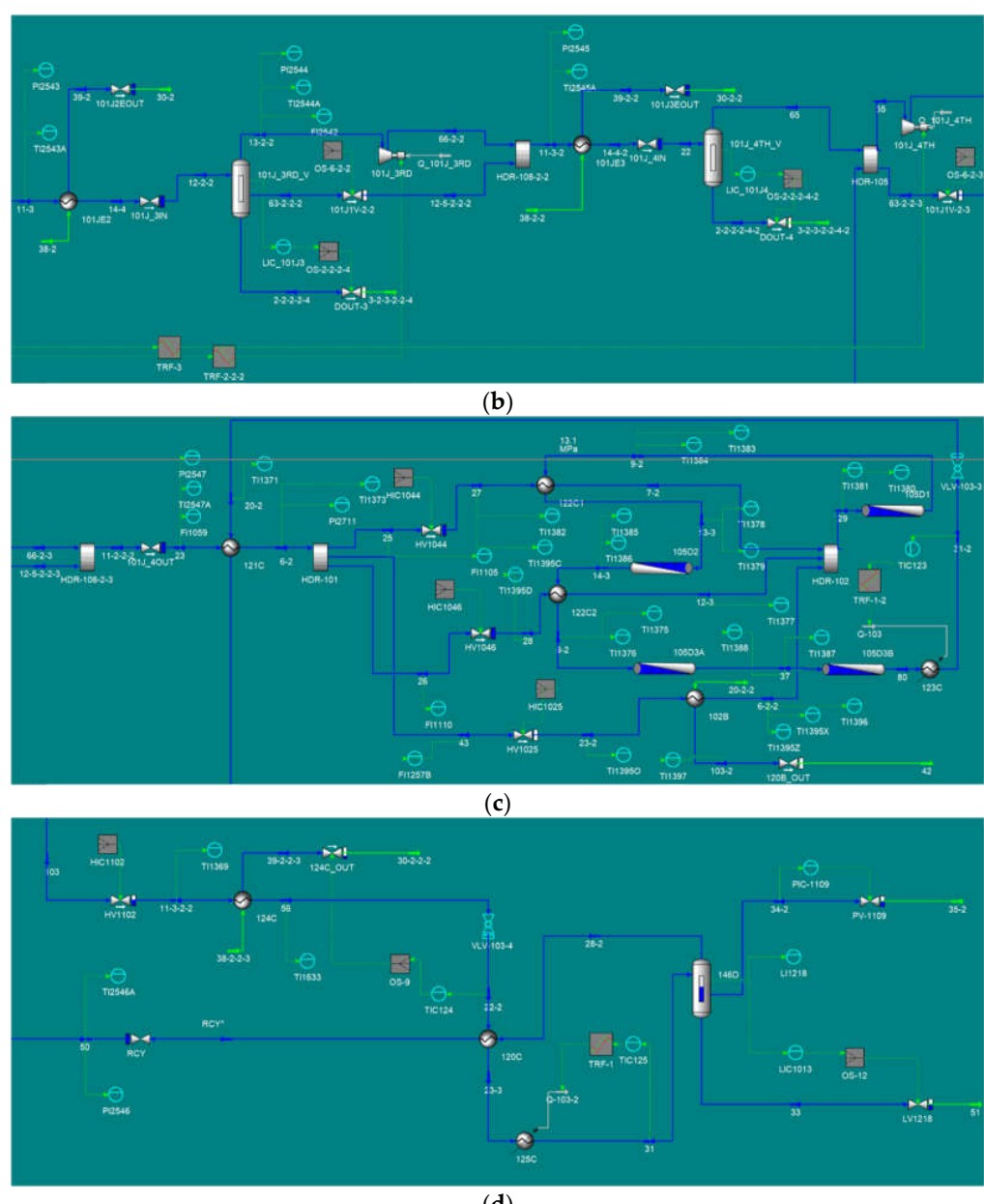

**Figure 3.** UniSim model of the green ammonia process: (**a**) syngas buffer section-1; (**b**) syngas buffer section-2; (**c**) reaction section; and (**d**) product refining section. The recycle logistics in (**d**) are connected to (**b**).

The buffer section's function was to pressurize syngas as required for the reaction; the synthetic ammonia section was a simulation of the synthetic ammonia reactor. The ammonia produced was separated from other products in the refining section, and unreacted syngas was recycled.

### 3.1.1. Data Reconciliation

The syngas compression section affected only the syngas pressure; as there were no actual plant data to reconcile, we focused on the logistics of syngas entering the reaction plant (stream 23). The core of this model is the reaction section; therefore, the reaction export logistics (stream 80) was also taken as the optimization objective. Circulating gas (stream 50) and fly-off gas (stream 34-2) were also key logistics in synthetic ammonia. The following variables were summarized as optimization variables, and actual factory data were used to reconcile the model:

- 1st compressor pressure drop.
- 2nd compressor pressure drop.
- 3rd compressor pressure drop.
- 4th compressor pressure drop.
- 1st stage reactor temperature.
- 2nd stage reactor temperature.
- 3rd stage reactor temperature.
- 4th stage reactor temperature.
- Stream 31 temperature.
- Stream 31 pressure.

Some researchers have used UniSim's optimizer to design an optimization objective function [34] to make a model better conform to an actual situation. However, this study adopted a simpler optimization mode for data reconciliation [35]. Adjusting the above 10 variables made the temperature of streams 23, 35-2, 50, and 80 (denoted as $T_{23}$, $T_{35}$, $T_{50}$, and $T_{80}$, respectively), the pressure of streams 23, 35-2, 50, and 80 (denoted as $P_{23}$, $P_{35}$, $P_{50}$, and $P_{80}$, respectively), and the ammonia concentration of streams 23, 50 and 80 (denoted as $Y_{23}$, $Y_{50}$, and $Y_{80}$, respectively) consistent with the actual situation. As the design load of the model is not the same as the actual plant load, the error between the stream flow-rate and the measured value was not considered.

Table 2 presents a comparison of the concerned streams' simulation results and real values after data reconciliation. There was little difference between the pressures and temperatures of the target logistics and actual factory data, and relative errors were within 10%. In particular, ammonia content was close to actual production data, and the model was consistent with real process data after data reconciliation. Table 3 presents the final variables' values.

**Table 2.** Comparison of target values after data reconciliation.

| Items | Simulation Value | Real Value | Δ |
|:---:|:---:|:---:|:---:|
| $T_{23}$ | 180.6 | 195 | 7.38% |
| $T_{35}$ | 0.0 °C | 0.0 °C | 0.00% |
| $T_{50}$ | 36.8 | 38 | 5.79% |
| $T_{80}$ | 403.1 | 440 | 8.39% |
| $P_{23}$ | 14 MPa | 15.5 MPa | 9.68% |
| $P_{35}$ | 13.1 MPa | 14 MPa | 6.43% |
| $P_{50}$ | 13.1 MPa | 14 MPa | 6.43% |
| $P_{80}$ | 13.8 MPa | 15.2 MPa | 9.21% |
| $Y_{23}$ | 3.7% | 3.5% | 5.71% |
| $Y_{50}$ | 5.0% | 4.95% | 1.01% |
| $Y_{80}$ | 19% | 19.5% | 2.56% |

**Table 3.** Key model parameters.

| Parameter | Value |
|:---:|:---:|
| 1st compressor pressure drop | 1500 kPa |
| 2nd compressor pressure drop | 3480 kPa |
| 3rd compressor pressure drop | 6440 kPa |
| 4th compressor pressure drop | 650 kPa |
| Ammonia reactor temperature (R1/R2/R3/R4) | 450 °C/418 °C/399 °C/399 °C |
| Ammonia reactor pressure | $1.358 \times 10^4$ kPa |
| Feed temperature of synthetic ammonia reactor | 324.7 °C |
| Ammonia separating tower temperature | 0 °C |
| Ammonia separating tower pressure | $1.316 \times 10^4$ kPa |
| Recycled gas temperature | 36.80 °C |
| Recycled gas pressure | $1.31 \times 10^4$ kPa |

### 3.1.2. Model Result

The feed composition of syngas referred to the actual feed stream design of the plant. See Table 4 for details of the feed stream parameters.

**Table 4.** Feed stream parameters.

| Item | | Synthesis Gas |
|---|---|---|
| Mass fraction | H2 | 74.84 |
| | N2 | 24.96 |
| | Ar | 0.19 |
| | NH3 | 0 |
| | $H_2O$ | 0 |
| Temperature/°C | | 35 |
| Pressure/MPa | | 1.3 |
| Flow rate/kg·h$^{-1}$ | | 10680 |

The calculation results of the final product are shown in Table 5. There were significant differences in production output values due to different design conformance, which were not caused by model errors. Other product stream indicators were consistent with actual plant values.

**Table 5.** Comparison of simulated data and actual production data.

| Parameters | Real Data | Simulated Data |
|---|---|---|
| Production output (kg/h) | 62,539 | 8300 |
| Production temperature (°C) | 2.2 | 2.3 |
| Production pressure (kPa) | 1862 | 1864 |
| Product purity (Mole %) | 99.91% | 99.90% |

Next, simulation research was conducted for variable working conditions based on the steady-state model to analyze its energy consumption performance. Combined with analysis of the actual situation, the most common situation faced by the green ammonia process was that hydrogen supplied by the water electrolysis device was extremely unstable, and the process's operation load required significant adjustment. Therefore, it was necessary to build a dynamic model based on the steady-state model to study the performance of the transition state's dynamic process.

### 3.2. Dynamic Model

UniSim was switched to a dynamic model to add control points and device size data. The main automatic control points were added according to the actual control scheme of the factory, as shown in Table 6.

**Table 6.** PID main control parameters.

| Control Points | P | I |
|---|---|---|
| Synthesis gas pressure PIC-101 | 1.00 | 3.00 |
| Reactor output temperature TIC-123 | 0.01 | 8.0 |
| Heat inlet temperature of recycled gas heat exchanger TIC-124 | 1.00 | 3.00 |
| Feed temperature of ammonia separator TIC-125 | 0.5 | 3.00 |
| Pressure of ammonia separator PIC-1109 | 2.00 | 3.00 |
| Ammonia separation tower level LIC-1103 | 1.00 | 3.00 |
| Feed N2 flow rate FIC-N2 | 0.30 | 0.30 |
| Hydrogen and nitrogen mixture tank level LIC-MIX | 1.00 | 3.00 |

The dynamic model was verified to ensure its accuracy. Steady-state data were used to verify dynamic operation data. The hydrogen feed fluctuated slightly during the dynamic process, resulting in dynamic data and steady-state model data for comparison and verification. The results are shown in Figure 4, indicating that the dynamic model results, after operation stabilization, were consistent with steady-state model data, indicating that the dynamic model is accurate.

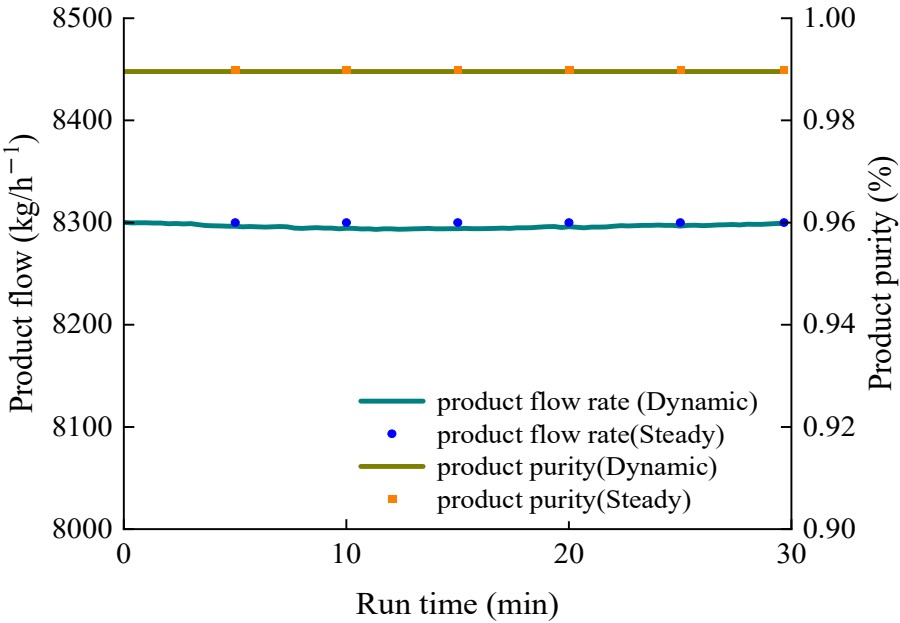

**Figure 4.** Dynamic model validation.

### 3.3. Transition State Process Simulation

Next, the verified model was used to simulate the transition state process under variable working conditions to analyze the process's energy consumption performance. The following four transition state processes were simulated according to the actual situation:

① Hydrogen feed decreased from 100% to 75%.
② Hydrogen feed reduced from 100% to 50%.
③ Hydrogen feed increased from 75% to 100%.
④ Hydrogen feed increased from 50% to 100%.

### 3.3.1. Transition State Process Product Curve

Transition state processes ① and ③ were simulated using dynamic models, and the influence of the transition state process on product quality and flow rate was determined. In all transition state processes, $H_2$ feed flow disturbance was added to the system after the system ran stably for 8 min.

According to Figure 5, the transition state process caused continuous flow fluctuation for approximately 110 min. When the working condition was adjusted (at 8 min), if the load was decreased, the purity of the product increased; if the load was increased, the purity of the product decreased. Product quality lasted approximately 10 min from change to stability. Continuous flow fluctuation was not conducive to the stable operation of the process device. It was noted that process control parameters should be optimized to minimize their influence on the device's stable operation to shorten fluctuation duration and oscillation amplitude.

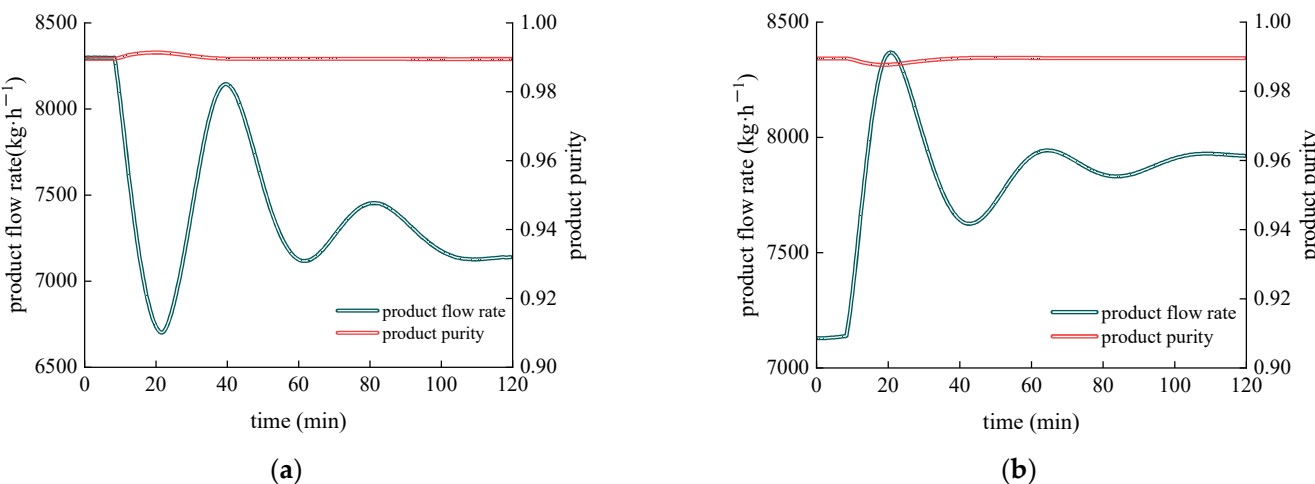

**Figure 5.** (**a**) Process ① product curve and (**b**) process ③ product curve.

### 3.3.2. Energy Consumption Curves of the Transition State Process

UniSim software was used to add dynamic disturbances to simulate the above transition state processes ①~④. Energy consumption curves are shown in Figure 6.

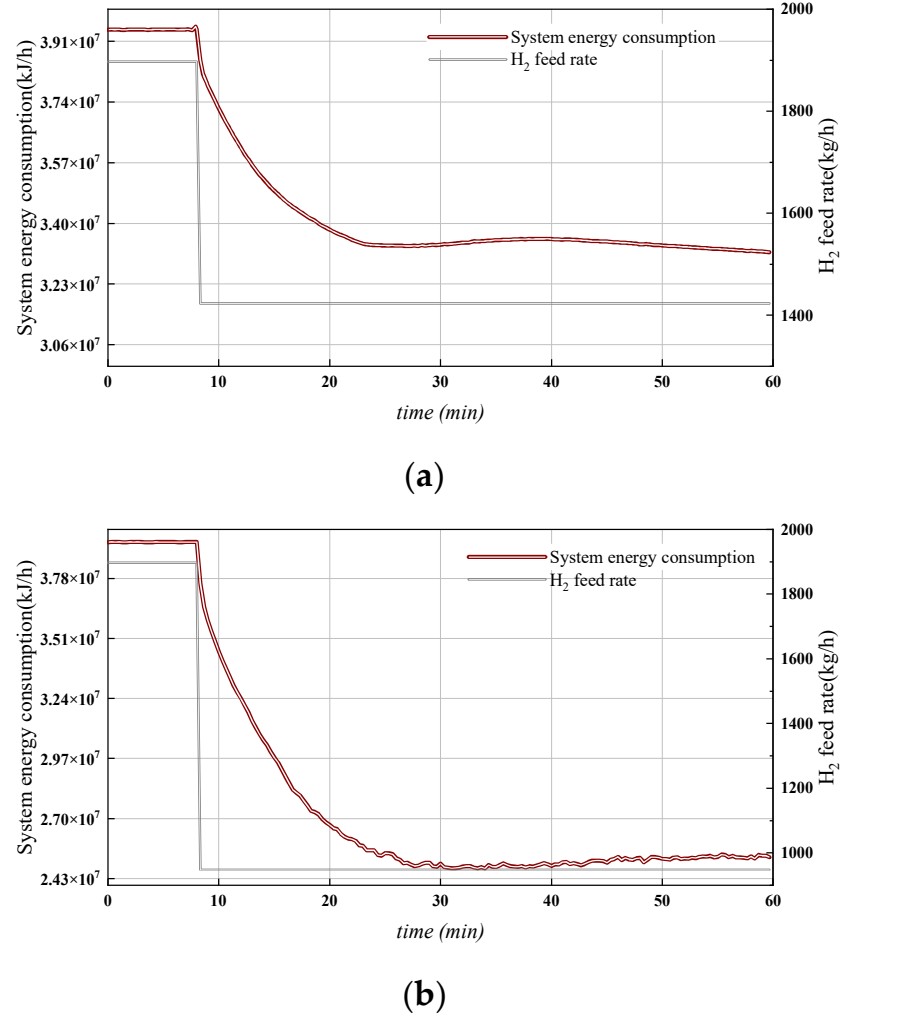

**Figure 6.** *Cont.*

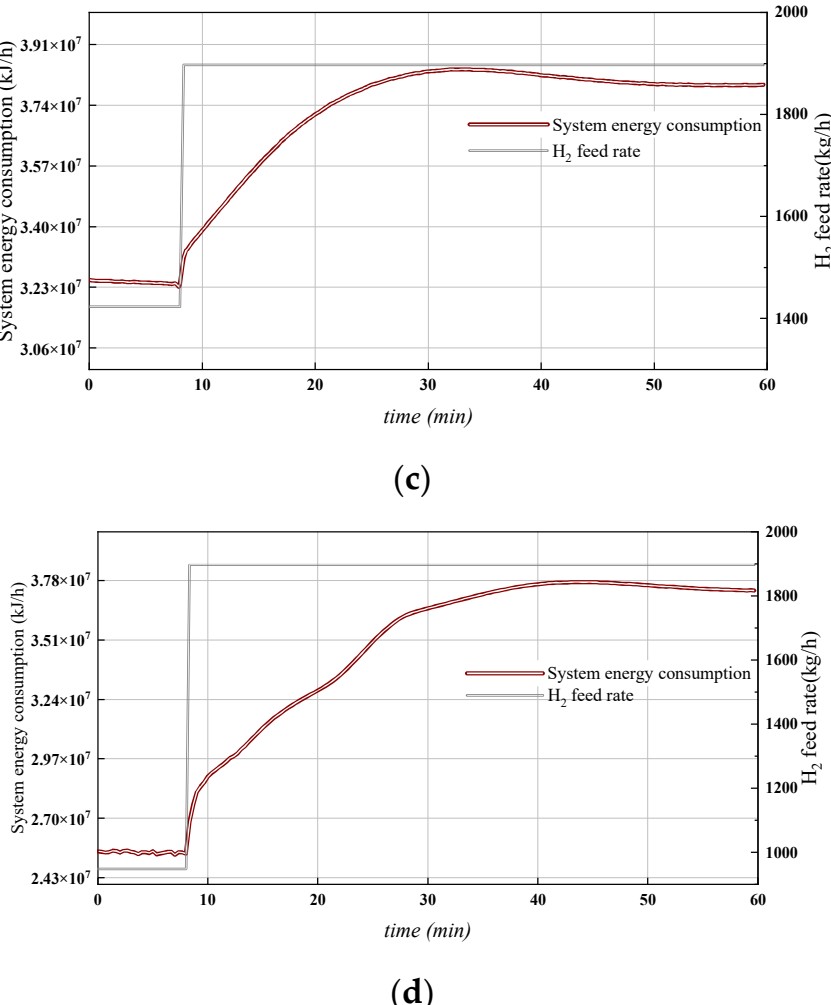

**Figure 6.** Energy consumption fluctuation curves caused by flow change in the transition state process: (**a**) process①energy consumption curve; (**b**) process②energy consumption curve; (**c**) process③energy consumption curve; and (**d**) process④energy consumption curve.

In all transition state processes, $H_2$ feed flow disturbance was added to the system after the system ran stably for 8 min. According to the analysis in Figure 6a, when the $H_2$ feed was reduced from 100% to 75%, the system's energy consumption gradually decreased. At 30 min, energy consumption gradually stabilized, and the system's energy consumption decreased by approximately 16%. Combined with the comparative analysis in Figure 6b, when the $H_2$ feed was reduced from 100% to 75%, the system's energy consumption significantly decreased and gradually stabilized at 30 min; energy consumption decreased by approximately 35.8%. This indicates that when the working condition was adjusted from large to small, the larger the adjustment range was, the more obvious the variation in the system's energy consumption range; however, the time difference in the system's energy consumption fluctuation was not obvious.

According to the analysis in Figure 6c, when the $H_2$ feed was increased from 75% to 100%, the system's energy consumption gradually increased. At 35 min, energy consumption gradually stabilized, and the system's energy consumption increased by approximately 15.4%. Combined with the comparative analysis in Figure 6d, when the $H_2$ feed increased from 50% to 100%, the system's energy consumption gradually increased and stabilized at approximately 50 min, and the system's energy consumption increased by approximately 32.1%. When the working condition was adjusted from small to large, the larger the adjustment range, the more obvious the system's change in energy consumption was. In contrast

to reducing the working condition, increasing the working condition required a longer energy consumption fluctuation time.

According to the above data analysis, the transition state process's fluctuation time was long under variable working conditions in the chloramine process, and the energy consumption for the four transition state processes fluctuated for more than 15 min. When the working condition increased, energy consumption fluctuated for a longer time. As the working condition of the green ammonia process frequently changed, it was not conducive to stable production. It was necessary to reduce the energy consumption fluctuation time and the energy consumption level and then optimize the process control parameters.

## 4. PID Parameter Optimization

### 4.1. PID Parameter Optimization Method Based on Variable Condition Energy Consumption Optimization

The entire PID optimization method framework is shown in Figure 7. Before PID parameter optimization, the optimization objective was to reduce energy consumption fluctuations and energy consumption levels so as to shorten the overall fluctuation time of the transition state. First, the unit's energy consumption was analyzed to determine optimal PID control parameters. The influence of changing each control point on the transition state process was analyzed using a comparative experiment so as to update the relevant PID control parameters, save energy, reduce consumption, and shorten the fluctuation duration.

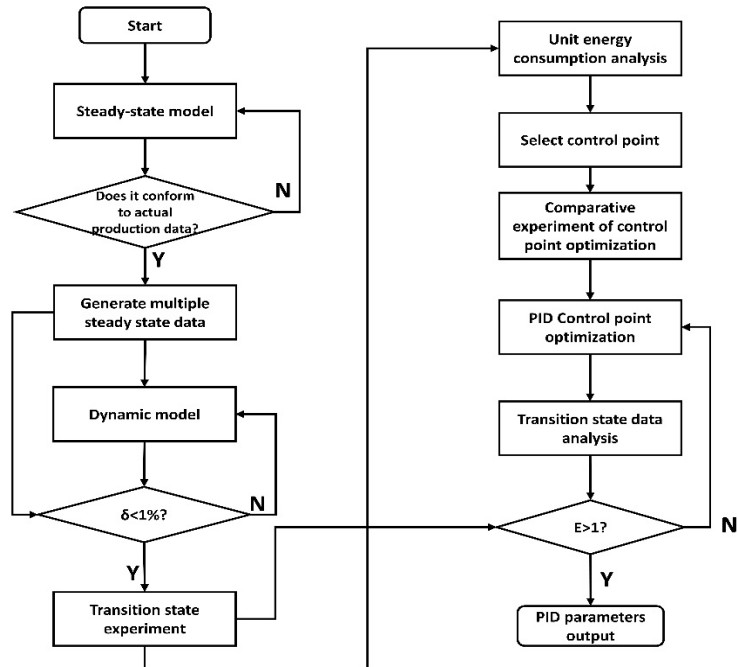

**Figure 7.** PID parameter optimization method framework based on energy consumption optimization under variable conditions. Y—yes and N—no.

The steady-state model was evaluated using the key parameter comparison method, as shown in Table 3. Dynamic model data were evaluated using parameter $\delta$.

$$\delta = \sum_{i=1}^{n} \left( \frac{\left| P_{ste,i} - P_{dyn,i} \right|}{P_{ste,i}} + \frac{\left| F_{ste,i} - F_{dyn,i} \right|}{F_{ste,i}} \right) \times \frac{1}{n} \times 100\% \tag{1}$$

where $P$ is product quality, $F$ is product flow rate, and $n$ is the number of selected points. The $\delta$ of the dynamic model was calculated as 0.1532% using Equation (1).

To quantify the optimization effect, parameter E was introduced to evaluate the optimization effect of energy consumption.

$$E = \frac{\int_{t_0}^{t_1} e(t)dt}{\int_{t_0}^{t_1} e_{op}(t)dt} \tag{2}$$

where $t_0$ is the starting point of the variable working condition, $t_1$ is the point when energy consumption is stable under the original process's parameters, $e(t)$ is energy consumption at time t, and $e_{op}(t)$ is energy consumption at time t after the PID parameter was optimized. Parameter E represents the influence of PID parameter adjustment on energy saving and consumption reduction in the process of variable load production fluctuation. The larger the value of E, the more obvious the optimization effect. This method is suitable for general process dynamic process parameter optimization and was used to optimize the PID parameters of the green ammonia process.

### 4.2. Unit Energy Consumption Analysis

The main equipment that consumed energy in the green ammonia process's stable operation included a heat exchanger and a compressor. In the dynamic green ammonia process model, Q101 represented the compressor's energy consumption, and $Q^{10}1J$-1st, $Q^{10}1J$-2nd, $Q^{10}1J$-3rd, and $Q^{10}1J$-4th represented the energy consumption of each stage of the four-stage compressor, respectively. Q103 represented the reactor outlet's heat exchanger energy consumption. Q103-2 represented the energy consumption of the ammonia separation tower's inlet heat exchanger.

First, the relationship between each energy consumption unit and total energy consumption was analyzed; dynamic model data for 100% load operation were selected for analysis. The results are shown in Figure 8.

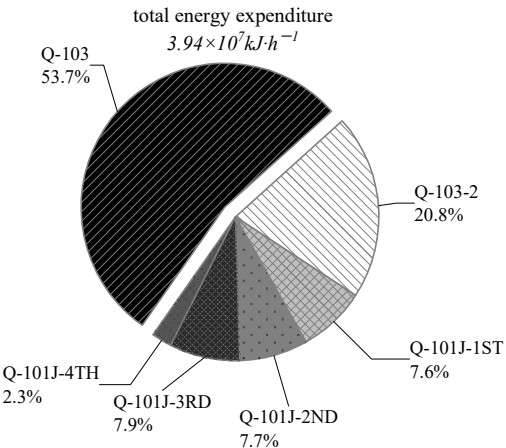

**Figure 8.** Energy consumption ratio of each unit.

According to the analysis of the unit energy consumption ratio illustrated in Figure 8, the total energy consumption of the green ammonia process was $3.94 \times 10^7$ kJ/h. The Q103 heat exchanger had the largest proportion of energy consumption, followed by the ammonia separation tower's inlet heat exchanger, Q103-2. These two heat exchangers account for approximately 75% of energy consumption. The energy consumption of the first, second and third-stage compressors was the same, accounting for approximately 8% of total energy consumption. The fourth stage compressor consumed the least energy.

Comparison data for the energy consumption of each unit before and after variable working conditions were analyzed as shown in Figure 9. Q103-2, the ammonia separation tower's inlet heat exchanger, had the largest energy consumption variation range of all units, and was the main cause of energy consumption fluctuation in the production process with variable working conditions. In addition, the first to third stage compressors, $Q^{10}1J$-1st,

$Q^{10}1J$-2nd, and $Q^{10}1J$-3rd, and the reactor outlet heat exchanger, Q103, had similar energy consumption variation ranges. Q103 and Q103-2 were the key optimization objectives.

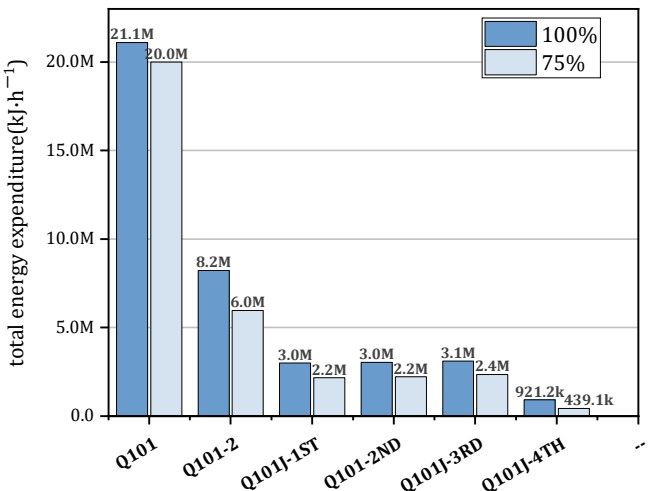

**Figure 9.** Comparison of unit energy consumption before and after variable working conditions.

### 4.3. Process Control Parameter Optimization

Combined with analysis of energy consumption units, Q103′s and Q103-2′s relevant PID controllers (TIC-123 and TIC-125, respectively) were optimized to reduce energy consumption levels and shorten fluctuation times. UniSim's PID controller adopted PI regulation [36,37], where P was the proportionality coefficient, and the value of P directly determined the stability of the controlled curve. The integral term I was introduced to eliminate residual error generated by proportionality regulation and to improve control accuracy. The value of P was important; when the value of P was reasonable, I was fine-tuned. Transition state process ① was selected to analyze the system's performance after process control parameters were adjusted.

First, parameter P of TIC-123 was optimized. Different P values were selected to simulate variable working conditions and to analyze dynamic energy consumption curves, as shown in Figure 10.

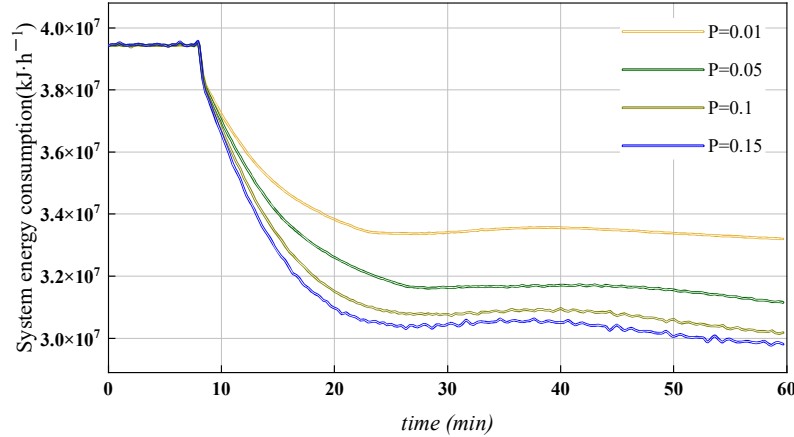

**Figure 10.** Comparison of different TIC-123 PID parameters' energy consumption curves.

When parameter P of the controller TIC-123 was continuously increased, it did not reduce the duration of energy consumption fluctuation, but it could reduce the energy consumption level after the variable working condition. When P increased to more than 0.15, there was a sawtooth fluctuation in the stationary phase of energy consumption; increasing P further was not conducive to smooth control.

Parameter P of TIC-125 was optimized; Figure 11 presents its energy consumption curves.

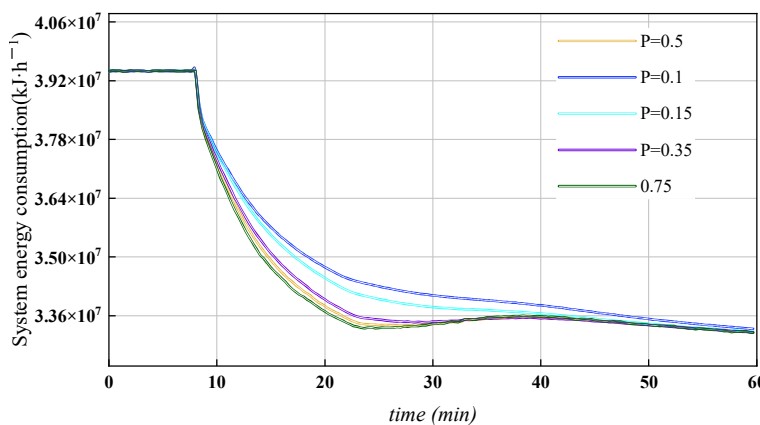

**Figure 11.** Comparison of different TIC-125 PID parameters' energy consumption curves.

Dynamic model simulations indicated that the maximum P value of TIC-125 should not exceed 0.8; otherwise, stable control could not be achieved. Therefore, five different P values within the 0.1–0.75 range were selected for simulation, the result is shown in Figure 11. Simulation results were reacted, and the parameter, P, of TIC-125 was reduced, which facilitated smoother control of the process's energy consumption fluctuation. It did not improve the energy consumption level.

To sum up, the parameter, P, of TIC-123 was selected as 0.1, and that of TIC-125 was selected as 0.15. Table 7 presents the final optimized PID control parameters after the integral parameter, I, was fine-tuned.

**Table 7.** Optimized PID parameters.

| Control Points | P | I |
|---|---|---|
| Synthesis gas pressure, PIC-101 | 1.50 | 3.00 |
| Reactor output temperature, TIC-123 | 0.1 | 10 |
| Heat inlet temperature of recycle gas heat exchanger, TIC-124 | 1.50 | 3.00 |
| Feed temperature of ammonia separator, TIC-125 | 0.15 | 5.00 |
| Pressure of ammonia separator, PIC-1109 | 3.00 | 4.50 |
| Ammonia separation tower level, LIC-1103 | 1.00 | 3.50 |
| Feed N2 flow rate, FIC-N2 | 0.50 | 0.50 |
| Hydrogen and nitrogen mixture tank level, LIC-MIX | 1.00 | 3.50 |

Transition state process ① was selected to analyze the system's performance after process control parameters were adjusted.

Compared with Figure 6a, Figure 12 shows that, after adjusting PID parameters, the system energy consumption stability time was approximately 25 min, which is approximately 5 min earlier than the adjustment premise, and energy consumption was reduced by a larger range. The total energy consumption was reduced by approximately 23.1%, which is higher than 16%.

According to Equation (2), $t_0$ was equal to 8, $t_1$ was equal to 30, the calculated value of parameter E was 1.05, and the energy consumption of the transition state process was optimized and improved.

Compared with Figure 5a, Figure 13 shows that the product flow reached a stable value at 100 min, 10 min sooner than prior to adjustment; the product flow decreased to 6200 kg/h, with no obvious difference between product purity before and after adjustment.

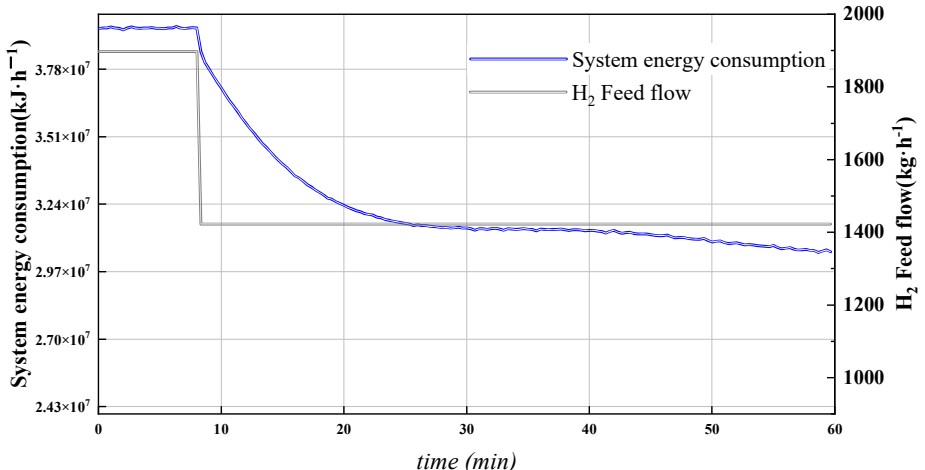

**Figure 12.** Process ① energy consumption curve after PID parameters were adjusted.

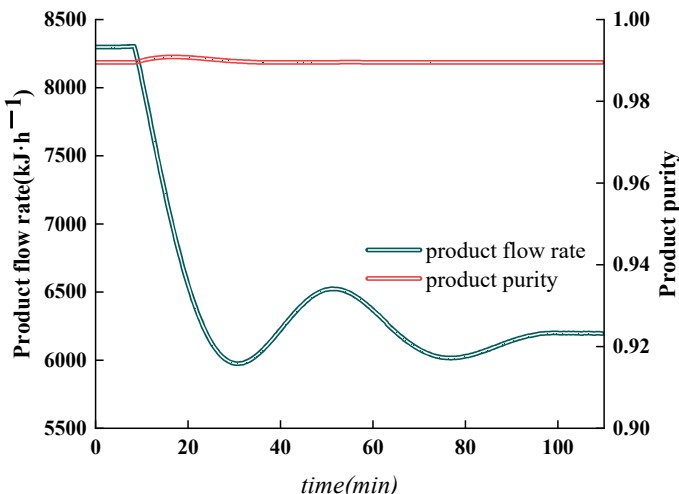

**Figure 13.** Process ① product curve after PID parameters were adjusted.

## 5. Conclusions

This study used the process route of a traditional ammonia plant to transform green ammonia. Thermal catalytic technology was adopted, hydrogen production from water electrolysis was used as the hydrogen source, reaction conditions were adjusted, a green ammonia process was designed, and the process flow model was built. The UniSim steady-state model was verified using the factory's actual production data, and based on this model, these data were transformed into a dynamic model to ensure their accuracy.

Based on the dynamic model, four transition state processes were simulated, and the product curves and energy consumption curves of transition state processes were analyzed. According to the proposed PID parameter optimization method (based on energy consumption optimization under variable conditions), PID control parameters were adjusted to shorten the flow fluctuation time of the transition state process and reduce energy consumption. According to this model experiment, the following conclusions can be drawn:

1.  Analysis of the transition state process of green ammonia showed that the flow rate of the product fluctuates for a long time, but the large changes were mainly concentrated during the 30 min after feed adjustment. During load adjustment, the energy consumption fluctuation time was 18–20 min, and the impact was greater when the load changed from small to large. Therefore, it is recommended that the feed be adjusted more slowly when the load is increased.

2. According to the analysis of energy consumption under variable conditions, the main energy consumption optimization of the green ammonia process was focused on the reactor's outlet heat exchanger and the ammonia separation tower's heat exchanger, which were the main points of energy consumption optimization.

3. Adjusting the PID parameter shortened the energy consumption fluctuation time of the transition state and reduced the energy consumption level; however, doing so sacrificed product output to a certain extent.

**Author Contributions:** Conceptualization, W.D. and X.L.; methodology and software, H.Z. and C.H.; validation and investigation, W.D.; data curation, C.H. and Y.D.; writing—original draft preparation, W.D. and Y.D. All authors have read and agreed to the published version of the manuscript.

**Funding:** The authors are grateful for the support of the National Key Research and Development Program of China (2021YFB4000502).

**Institutional Review Board Statement:** Not applicable.

**Informed Consent Statement:** Not applicable.

**Data Availability Statement:** Data presented in this study are available on request from the corresponding author. Data are not publicly available due to privacy concerns.

**Conflicts of Interest:** The authors declare no conflict of interest.

## Nomenclature

| | |
|---|---|
| $\delta$ | Parameter to evaluate dynamic model. |
| $P_{ste,i}$ | Steady-state model product quality. |
| $P_{dyn,i}$ | Dynamic-state model product quality. |
| $F_{ste,i}$ | Steady-state model product flow rate. |
| $F_{dyn,i}$ | Dynamic-state model product flow rate. |
| E | Parameter that represents the optimization effect. |
| $E$ | Energy consumption, $kJ \cdot h^{-1}$. |
| $e_{op}$ | Optimized energy consumption, $kJ \cdot h^{-1}$. |
| $t_0$ | Start time of energy consumption fluctuation, min. |
| $t_1$ | End time of power fluctuation, min.[1] |

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
