# Peer review of "Dynamic Simulation Analysis and Optimization of Green Ammonia Production Process under Transition State"

_processes, doi:10.3390/pr10102143_

Round 1

Reviewer 1 Report

The manuscript shows good features in terms of originality of concepts, technical soundness, importance of results, clarity of presentation, and awareness of the literature. The text is generally tidy, but some aspects need revision. The various issues to clarify, correct and improve are listed below.

·       Figure 3 is illegible. Please improve it a lot.

·       Eq.1. eq. n is in lower case, but in capital letter in the text

·       Eq.2. Why integrals? a discretized equation is here awaited.

·       A general revision of the text is suggested. English is not always fluent.

·       A typo. Page 10/16: “for general process dynamic process parameter optimization”. A double process is used.

·       When the authors discuss at lines 95-99 the fact the using a process simulator as Unisim has evidenced possibilities of energy efficiency and optimal design the authors should consider that recent methods have been proposed to overcome the burden linked to simulation costs using analytical simplifications. An example of such a methodology applied to a very common and critical distillation unit can be found “Analytical RTO for a Critical Distillation Process Based on Offline Rigorous Simulation.” (2022).

·       Tab 4. The authors should better explain which variables are used for the data reconciliation with the actual plant. Moreover, the authors should consider and at least mention that optimization strategies have been used in literature for exploiting such activity and obtain discrepancies less than the one shown in Table 4 for the flow rate of production output. See e.g.:

"A rigorous simulation model of geothermal power plants for emission control" (2020) for an example on data reconciliation for a model of a geothermal power plant, with a rigorous simulator

“Implementation of an Industry 4.0 system to optimally manage chemical plant operation (2020) and “Optimally Managing Chemical Plant Operations: An Example Oriented by Industry 4.0 Paradigms (2021)” for an example of a digital twin/RTO platform of a chemical plant facility with similar issues of data reconciliation.

Reviewer 2 Report

The text is in many places difficult to read and the English language should be drastically improved.

The present text was not carefully enough prepared: subscripts and underscripts are missing in many places

All acronyms should be defined before use. HB stands properly for Haber-Bosch but it is not defined.

What is the definition of green ammonia ?

How is defined the load of 62539 kg/h ?

Proper reference is need for UniSim.

Line 120: why was the Peng-Robinson equation selected ?

Figure 3: is is absolutely non readable

Line 156: this sentence does not have a verb.

Table 4: is three digits reasonable for a temperature, even when it is calculated ?

Figure 6: yellow lines are not readable.

Reviewer 3 Report

Reviewer comments:

 Good and valuable paper ready for publication after some corrections that authors have to accomplish.

 ·        On the end of the introduction section, a structure of the work should be briefly presented, so that the reader can get an idea of what is presented in the work;

·        For a better appearance, a 12 pt space must be kept between the figure and the text, valid before and after the figure; This rule must be applicable also for the tables;

·        It would be better for the figures to be positioned in the center of the page for a better page appearance. The journal formatting rules for authors should be consulted;

·        Figure 3 exceeds the page edges and must be adjusted so that it can be placed within the printable limits of the page; Same situation for figure 6;

·        Tables 2, 3 and 5 also exceed the printable page limit and must be re-sized to fit on the page;

·        Figure 10 title must be on the same page as the figure;

·        The font used for the values and text presented in the graphic representations (e.g. figure 12 ) should be standardized in order to obtain a better visibility for those presented;

By re-organizing the work it is possible to avoid using the last page occupied only with a few bibliographic references.

Round 2

Reviewer 1 Report

The authors have answered in detail to all my concerns and issues raised in the first round of revision. As residual concern, to enhance references on data reconciliation, as said in my first round of revision, authors are suggested including the following work: “Optimally Managing Chemical Plant Operations: An Example Oriented by Industry 4.0 Paradigms (2021)”.

Author Response

Thank you for your advice. I have cited this paper in the place of data reconciliation, and it is the 35th reference of this paper.

Reviewer 2 Report

The new text is much better. Just remove Marco line 136. Only family names should be used.

Author Response

Thanks for your proposal. I have removed Macro from the newly submitted manuscript.